# Efficacy of Early and Enhanced Respiratory Physiotherapy and Mobilization after On-Pump Cardiac Surgery: A Prospective Randomized Controlled Trial

**DOI:** 10.3390/healthcare9121735

**Published:** 2021-12-15

**Authors:** Georgios Afxonidis, Dimitrios V. Moysidis, Andreas S. Papazoglou, Christos Tsagkaris, Anna Loudovikou, Georgios Tagarakis, Georgios T. Karapanagiotidis, Ioannis A. Alexiou, Christophoros Foroulis, Kyriakos Anastasiadis

**Affiliations:** 1Cardiothoracic Surgery Department, General University Hospital of Larissa, University of Thessaly, 41110 Larissa, Greece; afxonidis.g@gmail.com; 2Cardiothoracic Surgery Department, AHEPA University Hospital, 53636 Thessaloniki, Greece; gtagarakis@gmail.com (G.T.); gkarapan@auth.gr (G.T.K.); drjannis@yahoo.com (I.A.A.); foroulis@auth.gr (C.F.); anastasi@med.auth.gr (K.A.); 3School of Medicine, Faculty of Health Sciences, Aristotle University of Thessaloniki, 54124 Thessaloniki, Greece; anpapazoglou@yahoo.com (A.S.P.); loudovikou.anna@gmail.com (A.L.); 4Athens Naval Hospital, 15561 Athens, Greece; 5School of Medicine, Faculty of Health Sciences, University of Crete, 71003 Heraklion, Greece; chriss20x@gmail.com; 6European Student Think Tank, Public Health and Policy Working Group, 1058 Amsterdam, The Netherlands

**Keywords:** physiotherapy, open heart surgery, early mobilization, enhanced physiotherapy, coronary artery bypass grafting, early physiotherapy, randomized-controlled trial

## Abstract

Background: This randomized controlled trial aimed to investigate the influence of physical activity and respiratory physiotherapy on zero postoperative day on clinical, hemodynamic and respiratory parameters of patients undergoing cardiac surgeries under extracorporeal circulation. Methods: 78 patients undergoing coronary artery bypass graft (CABG) or/and valvular heart disease surgeries were randomly assigned into an early and enhanced physiotherapy care group (EEPC group; n = 39) and a conventional physiotherapy care group (CPC group; n = 39). Treatment protocol for the EEPC group included ≤3 Mets of physical activity and respiratory physiotherapy on zero post-operative day and an extra physiotherapy session during the first three post-operative days, whereas the CPC group was treated with usual physiotherapy care after the first post-operative day. The length of hospital and intensive care unit (ICU) stay were set as the primary study outcomes, while pre- and post-intervention measurements were also performed to assess the oxymetric and hemodynamic influence of early mobilization and physiotherapy. Results: Participants’ mean age was 51.9 ± 13.8 years. Of them 48 (61.5%) underwent CABG. Baseline and peri-procedural characteristics did not differ between the two groups. The total duration of hospital and ICU stay were significantly higher in the CPC group compared to the EEPC group (8.1 ± 0.4 days versus 8.9 ± 0.6 days and 25.4 ± 3 h versus 23.2 ± 0.6 h, *p* < 0.001, respectively). Statistically significant differences in pre-intervention oxygen saturation, and post-intervention PO2 and lactate levels were also observed between the two groups (*p* = 0.022, 0.027 and 0.001, respectively). Conclusion: In on-pump cardiac surgery, early and enhanced post-procedural physical activity (≤3 METS) can prevent a prolonged ICU stay and decrease the duration of hospitalization while ameliorating post-operative hemodynamic and oxymetric parameters.

## 1. Introduction

Millions of patients benefit annually from open heart surgery, while the actual number of such operations in western countries remains stable over the last decade, despite the current advancements in cardiovascular medicine, offering pharmacological or minimally invasive alternatives [1,2]. However, open heart surgeries are still associated with increased risk for cardiovascular morbidity, respiratory complications and prolonged length of intensive care unit (ICU) stay and hospitalization, which increases the financial healthcare burden [3]. Concurrently, physiotherapy and early physical activity have been considered as crucial factors with a seemingly positive impact on post-procedural functional capacity, muscle weakness, and prevention or mitigation of post-operative complications succeeding heart surgery, thereby leading to quality of life amelioration [4].

Nevertheless, despite the apparent sequelae of reduced mobility and subsequent muscle weakness in patients undergoing open heart surgery, the effect of the extent and timing of early respiratory physiotherapy and mobilization on the length of ICU stay in cardiothoracic patients has not been thoroughly investigated [5]. Furthermore, significant variability exists in physiotherapy strategies and early mobilization practices [6]. Several studies have attempted to examine whether physiotherapy and rehabilitation might be an effective non-pharmacological tool to improve post-operative outcomes and reduce hospitalization stay [7,8,9,10,11]. Nonetheless, none of them investigated the influence of intensified/intensive post-operative physiotherapy and rapid mobilization.

On the basis of the above, we conducted a randomized-controlled trial to investigate the influence of enhanced post-operative physiotherapy and mobilization on patients undergoing open heart surgery. In particular, we aimed to inquire into the potential impact of early mobilization and physical activity combined with extra sessions of active post-operative physiotherapy, on the duration of ICU stay and hospitalization, as well as on clinical and laboratory parameters following open heart surgery.

## 2. Materials and Methods

### 2.1. Study Design and Sample Size Calculation

A double-blinded randomized clinical trial was designed to investigate the effect of respiratory physiotherapy and early mobilization and exercise on patients undergoing cardiac surgery [coronary artery bypass grafting (CABG) or valvular surgery] under extracorporeal circulation. The protocol of the study and all trial procedures conformed to the CONSORT Statement recommendations for reporting randomized trials [12].

Based on a minimum effect size of interest approach to calculate the sample size required to detect a statistically significant difference between the intervention group and the control group in terms of the primary outcome, 80 study participants (40 in each group) were required in order to have a 90% power to detect the effect at a significance level of 0.05.

### 2.2. Study Participants and Eligibility Criteria

Patients admitted to the University Cardiothoracic Surgical Departments of the University Hospitals of Thessaly and Thessaloniki between February 2019 and December 2020 to undergo elective CABG or valvular surgery with cardiopulmonary bypass were randomly allocated to the conventional physiotherapy care (CPC) group or the early and enhanced physiotherapy care (EEPC) group to be included in this trial.

Patients of both genders were invited to participate in this study without any race limitation. The inclusion criteria consisted of (1) a written informed consent to participate in the study, (2) a Glasgow Coma Scale score = 15, and (3) musculoskeletal and cardiopulmonary conditions suitable for the accomplishment of the proposed activities. On the contrary, patients were excluded from the study if they met one or more of the following criteria: (1) emergency, non-elective cardiac surgery, (2) hemodynamic instability preventing protocol performance, (3) breathing discomfort or invasive ventilator support or oxygen saturation below 90%, (4) severe neurological sequelae or neurodegenerative disorders, and (5) any mobile disability that did not allow them to perform exercise according to our protocol.

### 2.3. Random Allocation and Blinding

Patients were randomly assigned to an EEPC (n = 39) or to a CPC (n = 39) by simple randomization. Eligible participants were allocated in each group through computer-generated random numbers (Research Randomizer Software). The anaesthesia, peri- and post-surgical management procedures were standardized for all patients (ECMO: Medtronic Performer CPB, Oxygenation System: Medtronic Affinity NT, Autotransfusion System: Medtronic Autolog IQ, Heater-Cooler System: Stockert Basic Heater-Cooler Unit, Online Monitoring System: Spectrum Medical System M, Intra-Aortic Balloon Pump: Maquet Datascope GS300, Blood Flow Meter: Medistim MiraQ™ system, post-excubation: Venturi mask 50% oxygen flow rate) and were performed by the same team of Cardio-thoracic surgeons and Emergency and Critical Care technicians in the cardiothoracic surgery department of the general university hospital of Larissa.

The study was a double-blinded trial so that the participants were unaware of the existence of the other group. The blinded researcher was also not aware whether the patients were in the CPC group or in the EEPC group. After collecting the data, the blinded researcher measured the defined outcomes of interest.

### 2.4. Intervention

Usual physiotherapy care was applied to participants of the CPC group twice a day commencing on the first post-operative day until their discharge. On the contrary, patients allocated to the EEPC group were provided with an extra physiotherapy session during first three post-operative days or until ICU discharge (three sessions per day). Except for the aforementioned enhanced physiotherapy strategy, patients of the EEPC group also received an early physiotherapy session performed during zero post-operative day in the ICU. However, there was no procedural difference in the physiotherapy sessions between the two groups. Both intervention groups received:Respiratory physiotherapy including guided deep breathing techniques, incentive spirometry exercises with the use of TriFlo Inspiratory Exerciser (Manufacturer: Plasti-Med Plastic Medical Products), chest percussion, a chest binder or the use of passive assistance from the physical therapist. Patients were also instructed to cough out, accompanied by moderate intensity chest percussion as per requirement.Physical activity which consisted of supported sitting over the edge of bed at bedside, performed on the first hours after extubation, assisted by the physiotherapist. Subsequently, patients were mobilized out of bed standing for one to two minutes and 10 to 50 steps of static walk, along with deep breathing exercises, followed by sitting on the chair beside the bed for half an hour, thereby achieving an estimated energy expenditure of no more than 3 metabolic equivalents (METs) of physical activity. The limit of 3 METs has been used in trials as it constitutes a borderline between sedentary and moderate-vigorous physical activity [7].

Briefly, the EEPC intervention comprised of an early mobilization and physiotherapy strategy during post-operative day zero and an extra-supplementary physiotherapy session taking place in the afternoon of the first three post-operative days or until ICU discharge. There was no difference in physiotherapy techniques and duration between the two groups. All physiotherapy interventions were undertaken by trained physiotherapists who were unaware of the existence of the two study groups and were also assisted by nurses and doctors of the ICU.

### 2.5. Ethical Considerations

The study protocol has been approved by the Institutional Review Board of Aristotle University of Thessaloniki, and the trial was conducted in compliance with the principles stated by the Declaration of Helsinki.

### 2.6. Recorded Variables and Measurements Performed

The measurements detailed in this section were taken before and after completion of the EEPC protocols on post-operative day zero. Despite no physiotherapy session being performed in the CPC group on post-operative day zero, the following hemodynamic and laboratory measurements, as well as clinical and demographic parameters, were also recorded for the CPC group, aiming to perform inter-group comparison analyses. After admission to the ICU, arterial and central venous and mixed venous blood samples were taken to accomplish these hemodynamic and laboratory measurements.

The demographics included data on gender, age, body mass index (BMI) and cardiac surgery performed. Hemodynamic measurements included the monitoring of body temperature, heart rate, respiratory rate, blood pressure, and oxygen saturation. Laboratory measurements consisted of serum electrolytes (Ca^2+^, Na^+^, K^+^), glucose and hemoglobin values as well as arterial blood gas analyses (pH, SvO2, PO2, PCO2, HCO3 and lactate). Echocardiographic and electrocardiographic assessment was also performed in every participant to record baseline left ventricular ejection fraction (LVEF) values via the Modified Simpson method (biplane method of disks) and to document the occurrence of any cardiac arrhythmia, respectively.

Further procedure-related data have been recorded to perform relevant inter-group comparisons. In particular, we documented: length of hospital stay (in days), length of ICU stay (in hours), duration of extracorporeal membrane oxygenation (ECMO) and aortic cross-clamp time (in minutes), number of coronary artery grafts as well as units of red blood cells (RBC) and fresh frozen plasma (FFP) transfused.

### 2.7. Definition of Study Outcomes

The primary outcome of this study was to identify any existing difference in the length of hospital and ICU stay between participants of the EEPC group and those of the CPC group. Secondary outcomes of interest were the comparisons of hemodynamic and laboratory measurements between the two study groups to detect any difference according to the physiotherapy protocol applied.

### 2.8. Statistical Analysis

Categorical variables are presented as frequencies with percentages, and continuous ones as means with standard deviations. Comparison among categorical variables was performed via the Pearson chi-square or Fisher’s exact test, whereas the Wilcoxon rank-sum or Student’s *t*-test were used to compare continuous variables, depending on the normality of data distributions.

Sample size estimation was performed using the G*Power tool. Thus, sample size estimation was based on the primary endpoint of the study and, in particular, on parameters affecting the hospitalization stay. For the F-test, the multivariate regression model using five predictors for increased hospitalization stay and assuming a power of 0.9, significance level of 0.05 and effect size of 0.3 (based on a preliminary analysis on the first 25 patients), the required number of subjects was 75. This initial sample size was intended to be increased by 15–20%, because of the high possibility that some patients might be lost to follow-up or would be eventually unsuitable for enrolment in our trial. Hence, we aimed for a total sample size of approximately 90 patients.

All analyses were conducted with the SPSS Statistics for Windows, Version 24.0 (Armonk, NY, USA: IBM Corp) and a two-sided *p* value of less than 0.05 was considered as statistically significant.

## 3. Results

After applying the inclusion and exclusion criteria, 78 patients (65 men, 13 women; mean age: 64.3 ± 8.9 years old) were finally included in this study among the 90 patients initially assessed for eligibility. Four patients died during surgery, three surgeries were cancelled, three surgeries were performed without the use of extracorporeal circulation, and two patients refused to participate in the study. Figure 1 describes the flowchart of patient participation; 39 patients were randomly allocated to EEPC and 39 others to the CPC. The mean number of physiotherapy sessions during the hospital stay was 12.3 ± 0.8 for the CPC group, and 16.6 ± 1.2 for the EEPC group, while the mean duration of physiotherapy sessions was 8.9 ± 2.1 min. EEPC was applied as expected in all 39 patients, and no physiotherapy session was cancelled or postponed in any of our randomized patients. No adverse events or complications were observed during physiotherapy sessions.

The baseline, pre- and post-operative data are summarized in Table 1, and show that after randomization, the two groups had similar characteristics regarding gender, age, and BMI. Surgical procedure data were also comparable between groups, demonstrated by similar ECMO time, aortic cross clamp time, type of surgery performed, number of coronary artery grafts and units of RBC and FFP transfused (all *p*-values > 0.05).

With regard to the primary outcome of this study, patients of the EEPC group had to stay in hospital for significantly less days than patients of the CPC group (8.1 ± 0.4 days versus 8.9 ± 0.6 days, mean difference: 0.8 days, 95% C.I.: 0.6–1 days, *t*-value = 7.39, Cohen’s d = 1.57, *p* < 0.001) after their surgery. The total duration of ICU stay was also significantly higher in the CPC group compared to the EEPC group (25.4 ± 3 h versus 23.2 ± 0.6 h, mean difference: 2.2 h, 95% C.I.: 1.3–3.2 h, *t*-value = 4.58, Cohen’s d = 1.02, *p* < 0.001). Sub-analysis on the length of hospital and ICU stay according to the type of surgery performed did not demonstrate any significant difference (ANOVA: *p* = 0.590 and 0.327, respectively).

The comparison of pre- and post-intervention laboratory, oxymetric and hemodynamic measurements between the two study groups is presented in Table 2 and shows that most of the measurements did not differ significantly. In particular, pre- and post-intervention sodium, potassium, calcium, glucose, and hemoglobin values were almost similar among the two groups (*p*-values > 0.05) with an existing non-significant trend towards higher glucose levels in the CPC group than those of the EEPC group (pre-intervention: 153.5 ± 39.1 mg/dL versus 140.6 ± 25 mg/dL, *p* = 0.087; and post-intervention: 156.8 ± 33.5 mg/dL versus 146 ± 24.1 mg/dL, *p* = 0.105). Analyses on the observed respiratory rates, blood pressure levels, body temperatures, venous oxygen saturation, arterial blood pH and PCO2 levels did not yield any significant differentiation (*p*-values > 0.05). However, mean arterial oxygen saturation was significantly higher in the CPC group prior to the intervention (99% versus 98.1%, *p* = 0.022), but non-significantly lower than that of the APG after the intervention (99% versus 99.4%, *p* = 0.128). Finally, mean PO2 levels were significantly higher in the EEPC group than in the CPC group after the intervention (192.5 ± 70.8 mmHg versus 159 ± 60.2, *p* = 0.027), whereas mean post-intervention lactate levels were significantly higher in the CPC group when compared to those in the EEPC group (1.5 ± 0.7 mmol/L versus 1.1 ± 0.2 mmol/L, *p* = 0.001).

## 4. Discussion

In this randomized-controlled trial enrolling patients undergoing selective open heart surgery, application of early and enhanced physiotherapy conferred a modest but significant advantage in the length of hospital and ICU stay over a conventional physiotherapy strategy. In addition, our study indicated a statistically significant difference in mean post-intervention PO2 between the EEPC and CPC groups in favour of EEPC. The main findings of this trial, as well as an illustration of our study’s protocol, including differences in physiotherapy strategy between groups, are depicted in Figure 2. Our study adds to the existing literature, as most of the studies dealt with enhanced sessions of physiotherapy after the first post-operative day or ICU discharge, or with enhanced pre-operative physical activity. To our knowledge, this is the first study to compare, in terms of hospitalization length, a conventional physiotherapy strategy with active physiotherapy, which included enhanced post-operative physical activity along with early mobilization.

Our results are consistent with those of other studies [8,9,13,14] indicating that early mobilization can decrease hospitalization length. Furthermore, a meta-analysis conducted by Y. Kanejima et al. suggested that early mobilization after cardiac surgery might improve physical function at discharge and subsequently prevent prolonged hospital stays [11]. For a patient undergoing open heart surgery, the days spent in the ICU, as well as the next three thereafter, constitute the most critical time of their post-operative phase [15]. Multiple organs, with lungs being of utmost importance, are prone to dysfunction during this period [16].

After open heart surgery, deterioration of functional capacity can be triggered by muscle weakness and proteolysis, which is induced by reduced mobility. Prolonged inactivity and muscle atrophy are responsible for atelectasis, the sensation of fatigue and aspiration pneumonia, which renders rehabilitation a ‘highly recommended’ healthcare strategy in the post-operative period of invasive cardiac procedures [17,18,19]. Therefore, the presence of a multi-professional team, including physiotherapists, in the ICU has been proven by several studies to contribute to early patient recovery, reduced need for mechanical ventilation support, and ultimately a fewer number of hospitalization days, through the prevention of respiratory complications [20].

On the other hand, some studies on post-operative ICU patients have questioned these outcomes and highlighted the absence of apparent differences between early mobilization and usual care [21,22] regarding the length of hospitalization. Furthermore, no study has indicated survival benefits for EEPC, and thus a decreased hospitalization length does not seem to be translated into lower rates of all-cause mortality or/and cardiovascular mortality [22]. This is also supported by the fact that the effects of EEPC are not reflected in changes in hemodynamic and laboratory indicators [7]. This is in harmony with our results, which only yielded a significant difference in post-operative PO2 and lactate levels between the EEPC and CPC groups.

Despite not having a clinically significant effect, in terms of hard clinical outcomes and laboratory parameters, enhanced and early physiotherapy could still play a crucial role in reducing healthcare costs and decongesting ICUs. Indeed, another study, with original data deriving from a large registry, also showed that early cardiac rehabilitation was associated with a lower length of ICU and hospital stay, and by such means significantly reduced costs [13]. As described above, our trial showed that a modest increase in physiotherapy sessions (approximately four sessions more than the control group) influences the length of hospitalization stay (mean difference between groups: 0.8 days), which features early and enhanced physiotherapy as a potently cost-effective intervention. This increase in physiotherapy sessions did not greatly affect everyday physiotherapy practice in our clinic, which might render this technique a rather applicable strategy. Besides, reducing costs by decreasing approximately one day of hospitalization stay could generate a positive feedback to physiotherapy protocol establishment and personnel enhancement. Also of note was that no adverse outcomes, such as oxygen desaturation, hypotension, acute coronary events, or arrhythmias were observed during protocol application. Given the above, our study protocol has a high level of feasibility, which encourages its application or the execution of similar strategies, and increases its external validity.

Our study should be interpreted in the context of its limitations. First of all, our randomized-controlled trial is single-centered with a relatively limited number of included patients. In the second instance, regarding our main outcome, the mean difference in hospitalization length of 0.8 days might be statistically significant, but constitutes a quite modest reduction in hospital stay, which prevents us from reaching definite conclusions. Additionally, we were not able to register our study protocol in a protocol database early, although all necessary approvals and certifications were given by the investigators’ institution. Furthermore, we were not capable of thoroughly elucidating our results on a pathophysiological basis, or to provide clinical explanations about the fact that decreased hospitalization and ICU stay was not attributable to altered hemodynamic and laboratory parameters. Moreover, we did not incorporate and analyze any techniques of pre-operative physiotherapy. Finally, our results demonstrated the effect of a combined intervention, which included both early mobilization and enhanced physiotherapy sessions. However, we were not able to separately assess the clinical weight and significance of each intervention on our outcomes. Future clinical studies could test different combinations of physiotherapy and mobilization activities to form the most cost-effective physiotherapy strategy after open heart surgery, which could optimally be based on the clinical parameters of each patient, thereby providing a personalized approach in cardiovascular physiotherapy.

## 5. Conclusions

In patients undergoing open heart surgery, early mobilization and physical activity, along with enhanced respiratory physiotherapy, significantly decreased the length of both ICU and hospitalization stay. However, these outcomes were not reflected in significant differences in post-intervention hemodynamic and laboratory parameters, except for increased PO2 and decreased lactate levels. Larger randomized, controlled trials are warranted to establish certain physiotherapy strategies and a structured physiotherapy program, which might lead to better clinical outcomes and faster postoperative recovery, decreasing the length of hospitalization and the subsequent financial burden.

## Figures and Tables

**Figure 1 healthcare-09-01735-f001:**
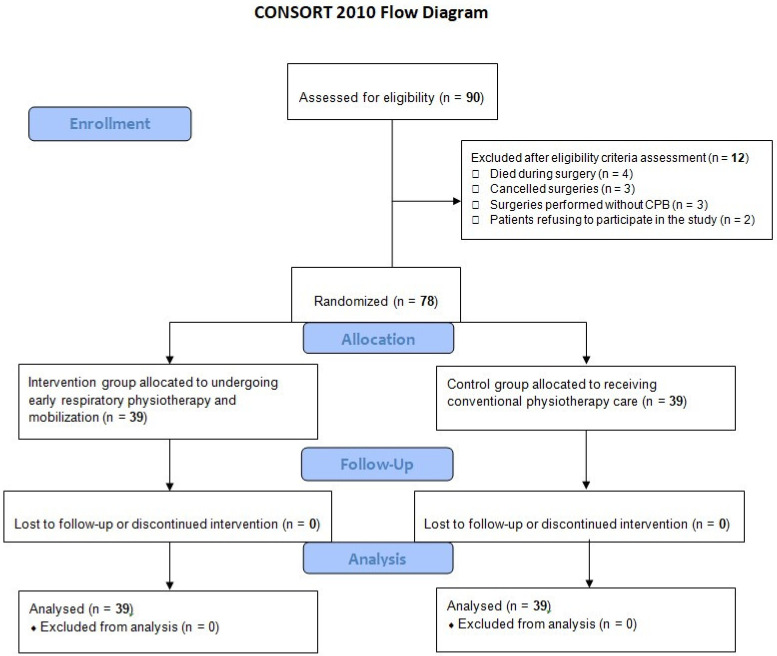
Flowchart of patient participation in this randomized controlled trial according to the CONSORT 2010 statement guidelines.

**Figure 2 healthcare-09-01735-f002:**
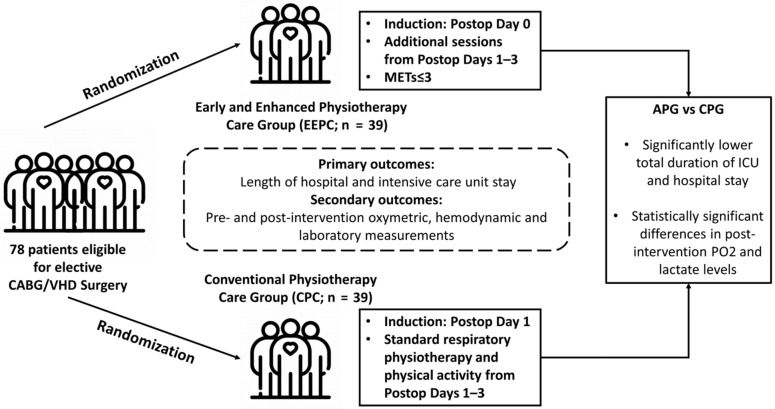
Study protocol and main findings of this randomized-controlled trial.

**Table 1 healthcare-09-01735-t001:** Baseline demographic and clinical characteristics of study participants.

Variable	EEPC Group (n = 39)	CPC Group (n = 39)	*p*-Value
Age (years)	63.5 ± 8.9	65.1 ± 8.9	0.424
Male gender, n (%)	34 (87.2%)	31 (79.5%)	0.362
Body mass index (kg/m^2^)	26.8 ± 4.2	27.9 ± 4.2	0.234
Left Ventricular Ejection Fraction (%)	50 ± 6	49.7 ± 6.5	0.856
Type of surgery:			
Coronary artery bypass graft, n (%)	25 (64.1%)	23 (59%)	
Aortic valve replacement, n (%)	9 (23.1%)	15 (38.5%)	
Mitral valve replacement, n (%)	5 (12.8%)	1 (2.5%)	0.119
Coronary artery grafts (number)	3.1 ± 0.8	3 ± 0.8	0.593
Red blood cells transfused (units)	0.9 ± 0.9	0.5 ± 0.7	0.063
Fresh frozen plasma transfused (units)	1.9 ± 1.1	1.9 ± 1.2	0.772
Cardiopulmonary bypass time (minutes)	100.1 ± 21.3	99 ± 27.7	0.848
Aortic cross-clamp time (minutes)	68.5 ± 19.1	69.7 ± 19.5	0.770

**Table 2 healthcare-09-01735-t002:** Comparison of pre- and post-intervention laboratory, oxymetric and hemodynamic measurements.

Variable	EEPC Group (n = 39)	CPC Group (n = 39)	*p*-Value	*t*-Value
Na^+^ (mEq/L):				
Pre-intervention Post-intervention	137.3 ± 2.6 137.6 ± 3	139.1 ± 6.6 137.6 ± 2.8	0.105 0.995	1.53 0.20
K^+^ (mmol/L):				
Pre-intervention Post-intervention	4.0 ± 0.4 4.1 ± 0.3	4.0 ± 0.7 4.1 ± 0.3	0.729 0.968	1.02 1.09
Ca^2+^ (mg/dL):				
Pre-intervention Post-intervention	0.9 ± 0.3 0.9 ± 0.2	0.9 ± 0.1 0.9 ± 0.1	0.293 0.488	1.01 0.84
Glucose (mg/dL):				
Pre-intervention Post-intervention	140.6 ± 25 146 ± 24.1	153.5 ± 39.1 156.8 ± 33.5	0.087 0.105	0.38 2.26
Hemoglobin (g/dL):				
Pre-intervention Post-intervention	9.6 ± 0.98 9.8 ± 1	9.6 ± 0.9 9.8 ± 0.9	0.813 0.972	0.24 −0.04
Respiratory rate (breaths per minute):				
Pre-intervention Post-intervention	23.4 ± 2.7 24.3 ± 2.8	24.3 ± 2.9 24.9 ± 1.7	0.183 0.263	1.34 1.13
Systolic arterial pressure (mmHg):				
Pre-intervention Post-intervention	130.4 ± 14.2 130.6 ± 15.2	127.1 ± 16 130.3 ± 15.7	0.329 0.942	−0.98 −0.08
Diastolic arterial pressure (mmHg):				
Pre-intervention Post-intervention	59.1 ± 8.5 59.8 ± 8	63.9 ± 19.8 59.3 ± 9.9	0.167 0.803	1.40 −0.25
Body temperature (°C):				
Pre-intervention Post-intervention	36.4 ± 2.3 37 ± 0.4	36.8 ± 0.4 36.9 ± 0.3	0.216 0.424	1.25 −0.80
Arterial oxygen saturation (%):				
Pre-intervention Post-intervention	98.1 ± 1.9 99.4 ± 1	99 ± 1.2 99 ± 1.2	0.022 0.128	2.34 −0.58
Venous oxygen saturation (%):				
Pre-intervention Post-intervention	61.8 ± 6.8 65.2 ± 6.9	58.4 ± 9 61.1 ± 9.4	0.187 0.141	−1.34 −1.59
Arterial blood pH (pH):				
Pre-intervention Post-intervention	7.4 ± 0.03 7.4 ± 0.04	7.4 ± 0.04 7.4 ± 0.04	0.954 0.951	0.47 0.52
PCO2 (mmHg):				
Pre-intervention Post-intervention	40.9 ± 4.3 40.6 ± 4.9	40 ± 4 40.9 ± 4.3	0.347 0.787	−1.61 0.27
PO2 (mmHg):				
Pre-intervention Post-intervention	137.5 ± 63.8 192.5 ± 70.8	149.4 ± 68.3 159 ± 60.2	0.430 0.027	0.61 −1.71
Lactate (mmol/L):				
Pre-intervention Post-intervention	1.5 ± 0.6 1.1 ± 0.2	1.6 ± 0.7 1.5 ± 0.7	0.694 0.001	1.04 2.25

## Data Availability

The data presented in this study are available on request from the corresponding author. The data are not publicly available due to privacy policy and ethical reasons.

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
