# Peer review of "Efficacy of Early and Enhanced Respiratory Physiotherapy and Mobilization after On-Pump Cardiac Surgery: A Prospective Randomized Controlled Trial"

_healthcare, 2021, doi:10.3390/healthcare9121735_

Round 1

Reviewer 1 Report

This work reported early mobilization and physical activity along with enhanced respiratory physiotherapy significantly decreased both ICU and hospitalization stay in patients undergoing open-heart surgery. Post-intervention parameters showed increased PO2 and decreased lactate levels in APC group. However, some statistical problems need to be confirmed.

  1. Post-intervention venous oxygen saturation seems to differ between APC and CPG groups.
  2. Please add P-value in the figures.
  3. The author may add one figure to illustrate the step of physiotherapy in APC group or difference between two groups.

Author Response

We wish to thank you for your detailed input on our work.

Post-intervention venous oxygen saturation seems to differ between APC and CPG groups.

Answer:

We would like to thank the Reviewer for this comment. Table 3 summarizes the outcomes of the comparison of pre- and post-intervention laboratory, oximetric and hemodynamic measurements. As shown, the mean post-intervention venous oxygen saturation in APC group (65.2±6.9%) was higher than in CPG group (61.1±9.4), but did not overpass the statistical significance threshold. On the contrary, as described in the manuscript, mean arterial oxygen saturation was significantly higher in the CPG prior to the intervention (99% versus 98.1%, p=0.022) but non-significantly lower than that of the APG after the intervention (99% versus 99.4%, p=0.128).

Please add P-value in the figures.

The author may also add one figure to illustrate the step of physiotherapy in APC group or difference between two groups.

Answer:

We would like to thank the reviewer for suggesting ways to strengthen the presentation of our results. P-values have now been added to the revised Figure 2.

We have also included an extra figure (Figure 3) which summarizes our study’s protocol, the differences in physiotherapy strategy between groups and our main findings.

Reviewer 2 Report

I would like to thank the editors of Healthcare for the opportunity to read the paper titled: Efficacy of early respiratory physiotherapy and mobilization after on-pump cardiac surgery: a prospective randomized con-trolled trial.

The authors present a randomized-controlled trial in which the effect on ICU and hospital stay of an enhanced mobilization and chest physiotherapy programme applied on the zero-postoperative day was compared to usual physiotherapy in open heart surgery patients. Secondary, they studied the effect of the proposed programme on clinical and laboratory parameters following open heart surgery.

The proposal is interesting and, as stated by authors, it adds to the existing literature about early physiotherapy and mobilization after surgery. The design seems appropriate to answer the aim of the study and the article is consistent in itself. Generally, impressions are good. However, several concerns about the methods and results need further detail/clarification for the manuscript to be published.

The title reflects the principal idea of the paper and is informative and explanatory in itself.  

In the introduction, the authors appropriately describe the background of the topic, from where the research question raised. This research question in clearly outlined.

In the methods section, why did authors base sample size on a minimum effect size? Please, indicate the exact value and give the rationale for this data.

How was randomization performed?

About the intervention, as described in the manuscript, differences between the control and the experimental treatments are not easy to follow. Please, the following clarifications are needed:

  • When chest physiotherapy is compared between treatments, it seems that techniques are the same, except for chest splitting and binder (“deep breathing exercises, coughing,… incentive spirometer, chest percussion” for CPG and “deep breathing techniques …incentive spirometer exercises … cough out accompanied by moderate intensity chest percussion” for APG). In that case, please indicate that the same techniques were used and mention the differences.
  • TriFlo Inspiratory Exerciser is a spirometer. As it is described in the manuscript, it seems that it is different from incentive spirometer exercises when it is really the device used to perform them. So rewritten is needed and for the CPG the device used should be also indicated. By the way, indicate the spirometer manufacturer as there are several.
  • Was the physical activity component of the APG based on previous research (i.e. Tarik et al. Effect of Early ≤ 3 Mets (Metabolic Equivalent of Tasks) of Physical Activity on Patient’s Outcome after Cardiac Surgery. J Coll Physicians Surg Pak 2017; 27: 490–494.)? In this case, please, indicate so.
  • Given the acute state of the participants, how were they controlled during the physical activity component? Which was the intensity target (unless you use a gas analyser, 3 METS are not measurable)?
  • How many physical therapists applied the intervention? How could they be blinded to the intervention group? This is an important flaw in physiotherapy studies.
  • As described in the manuscript, the control and the experimental treatments were applied twice a day. So, when authors say “the APG group, were provided with an extra physiotherapy session during first three post-operative days or until ICU discharge.”, what is intended to say if both groups received the same number of sessions per day?

Was the study protocol registered in a protocol database (i.e. clinical trials)? Please, indicate.

Regarding the statistical analyses, the authors indicate they have performed the Wilcoxon rank-sum or Student's t-test to make between-group comparisons. However, they did not indicate effect sizes for these tests. They should be included.

In results section, the authors repeat in the text data that is available in tables (means and ds, p value …). However, statistics (t, Z, d) are not indicated and they are not included in the tables. Please, include in the text the most relevant data, avoiding repetition (perhaps table 2 can be omitted).  I recommend to write them concisely.

I would recommend to include some results about the feasibility of the studied protocol (average no. of sessions and duration, if they could be applied twice a day as expected, adverse events or complications…).

In line with my previous comment, I would suggest to comment about the feasibility of the studied protocol in the discussion.

Round 2

Reviewer 1 Report

After revision, the author presented interesting teaching points clearly and improved the images. The study protocol was also clearly described. The work proved the importance of early respiratory physiotherapy in cardiac surgery. 

Author Response

We would like to thank the Reviewer for this constructive feedback and for helping us elaborate on raised concerns. His/her contribution to the improvement of our manuscript was remarkable.

Reviewer 2 Report

The authors present a randomized-controlled trial in which the effect on ICU and hospital stay of an enhanced mobilization and chest physiotherapy programme applied on the zero-postoperative day compared to usual physiotherapy in open heart surgery patients. Secondary, they studied the effect of the proposed programme on clinical and laboratory parameters following open heart surgery.

The proposal is interesting and, as stated by authors, it adds to the existing literature about the importance of early physiotherapy and mobilization after surgery. After the authors´ revision, the methodology of the study is easy to understand and the design is coherent with the aim of the study. Generally, impressions are good. I think it can be published in the reviewed form.

Author Response

(The authors gave the same response as above.)
